ecology/environmental science/developmental biology

*Tribolium*, sperm, reproduction, fertility, climate change, testes volume

**Authors for correspondence:**
Kris Sales
e-mail: krisreynardsales@gmail.com
Matthew J. G. Gage
e-mail: m.gage@uea.ac.uk

# Fertility and mortality impacts of thermal stress from experimental heatwaves on different life stages and their recovery in a model insect

Kris Sales, Ramakrishnan Vasudeva and
Matthew J. G. Gage

School of Biological Sciences, University of East Anglia, Norwich Research Park,
Norwich NR4 7TJ, UK

KS, 0000-0002-7568-2507; RV, 0000-0002-3831-0384;
MJGG, 0000-0003-3318-6879

With climate change creating a more volatile atmosphere, heatwaves that create thermal stress for living systems will become stronger and more frequent. Using the flour beetle *Tribolium castaneum*, we measure the impacts of thermal stress from experimental heatwaves in the laboratory on reproduction and survival across different insect life stages, and the extent and pace of any recovery. We exposed larvae, pupae, juvenile and mature adult male beetles to 5-day periods of heat stress where temperatures were maintained at either 40°C or 42°C, a few degrees above the 35°C optimum for this species' population productivity, and then measured survival and reproduction compared with controls at 30°C. Mortality due to thermal stress was greatest among juvenile life stages. Male reproductive function was specifically damaged by high temperatures, especially if experienced through pupal or immature life stages when complete sterility was shown at reproductive maturity; larval exposure did not damage adult male fertility. High temperatures impaired testis development and the production of viable sperm, with damage being strongest when experienced during pupal or juvenile adult stages. Despite this disruption, males recovered from heat stress and, depending on the stage of exposure, testis size, sperm production and fertility returned to normal 15–28 days after exposure. Our experiments reveal how thermal stress from heatwave conditions could impact on insect survival and reproduction across different life stages, and the potential and timescales of recovery.

# 1. Background

Increased atmospheric volatility under climate change will create more extreme weather events such as heatwaves, usually defined as at least five consecutive days where daily thermal maxima exceed the average local maximum by 5°C [1]. These environmental conditions are predicted to become increasingly widespread [1], frequent [2], intense [3] and prolonged [4]. Heatwaves generate unusually extreme conditions, often with short and unpredictable onsets, which can be particularly disruptive for biological function [5]. Ecologists, therefore, appreciate that such extreme climatic events may have stronger impacts on living systems than more gradual average changes [6,7]. Here, we measure the impacts of thermal stress associated with heatwave conditions on survival and reproduction in an insect model system.

The reproductive function has long been recognized to be especially sensitive to heat (e.g. [8]). In endotherms, male fertility can be compromised by environmentally relevant increases in temperature that are well below a species' critical limits [9–11]. Male mice (*Mus musculus*), for example, showed approximately 80% reductions in fertilization rates following exposure to an air temperature of 32°C for 24 h, compared with controls at 21°C [12]. Hyperthermia experiments with several endotherm models have revealed compromised reproductive function that resulted from declines in sperm counts [13], reduced motility [14], chromatin decondensation [15], DNA fragmentation [13], sperm abnormalities [10] and poor embryo or offspring viability [15]. This thermal sensitivity can explain why testes have evolved to occupy an external position in many mammal species, allowing spermatogenesis and sperm storage at 2–8°C below core body temperature [16].

Far less attention has been directed at understanding the impacts of thermal stress on reproduction in ectotherms. This is despite the most of biodiversity being ectothermic [17], and ectotherm physiology being more sensitive to ambient temperature changes [18,19]. The evidence so far indicates that male fertility can also be very sensitive to thermal stress in ectotherms (e.g. [20–24]), but knowledge lags significantly behind that of endotherms, and a greater understanding of 'thermal fertility limits' (TFLs) across different taxa in the context of climate change has been recently highlighted as a priority [25]. Accordingly, we examine how ectotherm reproduction and survival are impacted by experimental heatwave simulations using an insect model exposed to stressful temperatures of 40 and 42°C for five consecutive days, 5 to 7°C above the thermal optimum for population fitness in this system [26,27]. Temperature maxima of 42°C or more have been recorded in the natural environment across 103 countries [28].

We use the flour beetle *Tribolium castaneum* to explore how thermal stress acts across different endopterygote insect life stages, and then measure the extent and pace of any recovery. The *Tribolium* model is taxonomically [17], ecologically [19] and developmentally [29] representative of a very large number of insects. Recent large-scale, long-term field studies have highlighted major declines in insect and arthropod abundance, even across sites protected from habitat loss and pesticides. In particular, German nature reserves recorded a 75% decline in flying insect biomass over approximately 40 years [30], and Puerto Rico's Luquillo rainforest field site showed 10- to 60-fold declines in arthropod abundance over the past approximately 50 years, linked to a 2°C increase in average temperature maxima [31]. More recently, analyses across 66 bumblebee species throughout North America and Europe have revealed that extinctions or failed colonizations are specifically linked to 'areas where local temperatures more frequently exceed species' historical tolerances' [7].

Experimental research into how thermal extremes impact on adult reproduction in *T. castaneum* has shown that exposure of males and mated females to 5 days at 42°C effectively halved male fertility, with a second period of stress causing almost complete sterility [23]. Female reproduction, by contrast, was inherently resilient to thermal stress, but compromised via inseminated sperm in female storage being damaged by experimental heat exposure. The decline in male reproductive performance was associated with decreased sperm production with reduced viability and ability to gain critical female tract storage sites. In addition to direct effects, transgenerational damage to the reproductive fitness and lifespan of offspring was revealed if either sires or fertilizing sperm (within female storage) had experienced thermal stress [23].

Here, we address two important questions about how heatwave conditions could impact on insect systems: (i) whether different life stages vary in their sensitivity to thermal stress and (ii) whether recovery from reproductive damage after the stress can occur. Most insects have a complex holometabolic life cycle that passes through a series of larval stages, a pupal stage where metamorphosis takes place to produce the adult imago, and frequently a final maturing stage to reach the reproductive adult. Differences in morphology, physiology and behaviour between these stages, especially with regard to

reproduction, may render them more or less sensitive to the effects of thermal stress, but we understand very little about variations in vulnerability [22,32]. We, therefore, explore how experimentally stable heatwave conditions that reduce reproductive performance in mature adult beetles [23] impact across different life stages, examining larval, pupal and immature and mature adult stage sensitivities, measuring both survival and subsequent reproductive function in males. In addition to measuring life stage-specific impacts, we explore whether damaging effects are transient or permanent, and the timescales and magnitude of any reproductive recovery following thermal stress.

# 2. Material and methods

## 2.1. Insect maintenance

Experimental beetles were sourced from the outbred Krakow Super Strain (KSS) [33], maintained under standard conditions of a 16 L : 8 D photoperiod at $30 \pm 1°C$, and $60 \pm 5\%$ RH with ad libitum fodder consisting of plain white organic flour (Dove's Farm Foods Ltd, Berkshire, UK), powdered organic brewer's yeast (9 : 1 by volume) (ACROS organics, Belgium) and topped with organic rolled oats (Morning Foods Ltd, Cheshire, UK) [34]. Stock populations were maintained as non-overlapping generations in 1 l containers, renewed every 35 days once a full life cycle had completed. Each generation was reproduced by approximately 300 adults mating and ovipositing for 7 days before removal from the population using 850 µm sieves (Endecotts Ltd, London, UK).

## 2.2. Thermal exposure

Thermal stress at either 40 $(\pm 0.5)°C$ or 42 $(\pm 0.5)°C$ temperatures were applied to beetles for 5 days at 60 $(\pm 5)\%$ RH, creating conditions that were 5–7°C above the 35°C optimum for several life-history traits and overall population fitness and productivity in *T. castaneum* [23,27], and 10–12°C above the standard experimental stock maintenance temperature of 30°C. Heat exposure was applied to beetles in Petri dishes, or Eppendorf blocks stratified down the centre of Octagon 20 Eco (Brinsea Incubation Specialists, Somerset, UK) or A.B. Newlife 75 Mk4 (A.B. Incubators, Suffolk, UK) incubators. Parallel controls were kept in standard identical conditions but at 30°C, which is a widely accepted experimental temperature regime for *T. castaneum* (e.g. [35]). Randomization of position, daily rotations and periodic monitoring of temperature using calibrated mercury thermometers (G.H. Zeal, London, UK) limited potential microclimatic confounds in the incubators. After the 5-day exposure to heat stress or control conditions, all individuals were held in a common garden thermal environment for 24 h at 30°C before experimental treatment or assays. For survival and reproductive fitness assays, individuals at their relevant stages were maintained and exposed in groups of 20 individuals within 5 cm Petri dishes containing 10 g of fodder, while individuals for measuring testis and sperm impacts were held as individuals in perforated Eppendorf tubes containing 0.5 g of fodder.

## 2.3. Survival and reproductive fitness

To measure survival and reproductive fitness, unmated individuals were exposed to heat stress (40 to 42°C) or control (30°C) conditions for 5 days. After a further 24 h in common garden conditions for all at 30°C, post-treatment survival was scored as the number of individuals within the groups alive, indicated by coordinated motion. To assay reproductive fitness, individuals were monogamously paired with unmated members of the opposite sex from stock populations for 48 h in 4 ml perforated vials containing 0.5 g of fodder, followed by counts of offspring production. All individuals in the reproductive fitness assays were sexually mature, being 10 ($\pm2$) days of age post-eclosion. Females were identified with a small dot of correction fluid on the dorsal thorax [36]. After the 48 h mating opportunity, males were removed, and females isolated to oviposit in 5 cm Petri dishes containing 10 g of fresh fodder, typically across an oviposition period of two 10-day blocks. After female removal and transfer between each 10-day oviposition block, Petri dishes were incubated under standard 30°C conditions for 35 days, allowing offspring to develop into adults. Dishes were then frozen at −6°C and the total offspring number counted to score reproductive fitness. Thus, reproductive fitness was total adult offspring produced across up to 20 days of oviposition (in 10-day blocks) following 48 h of mating access. Twenty days of offspring production, following 2 days of mating opportunity, has previously been shown to account for over half of a female's subsequent reproductive fitness and

correlates significantly with total offspring production following such mating opportunities in *T. castaneum* ($R_s = 0.55$, $p = 0.001$, $n = 46$) [37].

## 2.4. Sex and life stage-specific assays

### 2.4.1. Sex-specific comparisons in the mature adult stage

To reconfirm the male-specific vulnerability of reproduction to thermal stress [23], we compared differences in survival and reproductive outputs (against 30°C controls) of pairs in which: (i) neither males or females had experienced heat stress, (ii) only males had been exposed to heat, or (iii) both males and females had experienced heat. Accordingly, $10 \pm 2$ days post-eclosion mature males and females (in single-sex groups) were exposed to either 30°C control or 42°C heat stress conditions for 5 days. After 24 h in a 30°C common garden, the within-group survival (containing $43.2 \pm 47.8$ s.d. individuals) was scored for control males ($n = 30$ groups), control females ($n = 39$ groups), heated males ($n = 45$ groups) and heated females ($n = 21$ groups) (electronic supplementary material, figure S1). Surviving males and females were monogamously paired to mate for 2 days in three treatments: (i) mating between 30°C control males and females ($n = 134$); (ii) mating between 42°C heated males and control 30°C females (male-specific effect; $n = 114$); and (iii) mating between 42°C heated males and females (male and female combined additive effect; $n = 33$) (electronic supplementary material, figure S1). Following the mating period, females were transferred to Petri dishes with fodder to oviposit within one 10-day time block under standard 30°C conditions, with reproductive fitness scored 35 days later.

### 2.4.2. Life stage-specific comparisons

To assay life stage-specific heat stress impacts, larvae, pupae, immature and mature adult stages were compared. Each life stage was sexed at the pupal stage [33], and stored as single-sex unmated groups in Petri dishes. The different life stages were then exposed to control (30°C) or heated (40°C or 42°C) conditions for 5 days. Two elevated temperatures were applied, as some earlier life stages displayed near-complete mortality at 42°C, and we aimed to measure heat stress conditions at different life stages on both survival and adult reproductive function. Survival for each group (containing $19.5 \pm 4.5$ s.d. individuals) was measured. Male survivors from each treatment developed in standard conditions following heat exposure to measure subsequent reproductive fitness at the sexually mature adult stage. Matings ran simultaneously across treatments to standardize male age. Treatment males were provided with 48 h mating opportunities with virgin KSS stock females in vials. Females were then isolated to oviposit for 20 days of offspring production (in two 10-day time blocks), and reproductive fitness (= adult offspring number) was counted 35 days later.

Timelines and treatments for the different life stage exposures are described in electronic supplementary material, figure S2, and below. Controls spent their entire development in standard 30°C conditions, and within-group survival ($n = 68$ groups) and reproductive fitness were measured (for a subsample of males, $n = 29$). Larvae were exposed to heat stress or control conditions as mixed-sex groups, with sex identified at pupation. Within-group survival was scored ($n_{40°C} = 9$, $n_{42°C} = 38$ groups), with males isolated as pupae then reproductive fitness measured as mature adults in parallel to the other treatments ($n = 19$). Heat exposure through the pupal life stage was applied to male groups ($n_{40°C} = 17$, $n_{42°C} = 5$ groups) for 5 days from their first day of pupation; subsequent reproductive fitness could not be measured because of high mortality. Heat stress exposure through the larval life stage was applied to final instar larvae 6 days prior to pupation. In the immature adult treatments, heat exposure was applied to males 3 days after pupal emergence, soon after the onset of cuticular hardening. Group survival was recorded ($n_{40°C} = 10$, $n_{42°C} = 22$ groups), then reproductive fitness of individuals ($n_{40°C} = 30$) was measured in parallel to the mature adult treatments. Mature male groups were exposed to 42°C thermal stress as $10 \pm 2$-day post-eclosion adults. Group survival ($n = 18$ groups) and individual reproductive fitness ($n = 30$) were measured.

## 2.5. Recovery assays of male reproductive fitness

### 2.5.1. General recovery assays

Male reproductive fitness recovery was tracked following single- and double-heat stress exposures to mature and immature adult male life stages. Males were periodically re-mated to control females with

increasing time following thermal treatment, and their subsequent reproductive fitness measured using the standard protocols described above. To standardize age relative to recovery time, in both control and heat stress male treatments, all females were sexed together in tandem with the males, so that male–female pairings involved adults of a similar age throughout. Timelines and treatments are described in electronic supplementary material, figure S3.

### 2.5.2. Mature adult exposure and recovery

Following heat stress exposure during the mature adult stage, the reproductive recovery of 10-day post-eclosion mature males was compared between 30°C control ($n = 35$) or 5-day 42°C heat stress ($n = 35$) treatments. After the treatment period, males were provided with mating and breeding opportunities with control females at 5, 10, 15 and 20 days following the heat stress. A second assay was conducted to investigate the recovery of males following exposure to two heat stress events, with an interim period of 10 days under control conditions at 30°C between the two. This double-heat stress exposure causes complete sterility in mature males [23]. Males were again exposed to either control 30°C conditions ($n = 63$), one ($n = 63$) or two 42°C ($n = 28$) heat stress conditions (electronic supplementary material, figure S3). Following exposure, males were provided with mating opportunities with control unmated females at 5, 10, 15 and 20 days after the control, single- or double-heat stress treatment period.

### 2.5.3. Immature adult exposure and recovery

A second assay explored whether adult males could recover from reduced reproductive fitness if they were exposed to heat stress during the immature stage (exposure began within 1 day of adult eclosion). Males were allocated to three treatments for comparison: (i) control 30°C conditions ($n = 19$), (ii) 42°C heat stress exposure as newly eclosed immature adults within 1 day of emergence ($n = 10$), or (iii) 42°C heat stress exposure as mature 10-day post-eclosion adults ($n = 16$) (electronic supplementary material, figure S2). As the assay focus was on whether immature adults could recover from heat stress, mating opportunities between control, immature and mature groups were synchronized relative to this heat exposure treatment, so that they occurred 1, 6 and 28 days after heat stress completion. As the mature adult treatment in the cohort was temporally out of synchrony with the immature stage, mature treatment males were provided with mating opportunities for comparison on days 1 and 23 following heat stress treatment.

## 2.6. Testis size and sperm recovery assays

### 2.6.1. Testis size

To investigate mechanistic damage to reproductive function, testis size, sperm counts and sperm viability assays were conducted. Prior to treatment and measurement, males were isolated singly in Eppendorf tubes to prevent any confounds arising from same-sex mating behaviour [36].

Two assays investigated the effect of heat stress exposure on testis size. The first compared testis sizes of mature males exposed to control conditions ($n = 16$), a single 5-day 42°C period of heat stress ($n = 16$), or two 5-day 42°C heat stress conditions split by a 10-day 30°C interval ($n = 17$) (electronic supplementary material, figure S3). The single-heat stress treatment ran simultaneously with the second heat stress period of the double-heat stress treatment. Following 24 h of maintenance under common garden 30°C conditions, all males were $31 \pm 2$ days post-eclosion when frozen at −20°C for subsequent dissection and testis measures (total $n = 49$ males).

The second assay explored the impact of heat stress conditions across different life stages on testis size at reproductive maturity. Comparisons were made between control and heat-exposed males for larvae ($n_{30°C} = 17$, $n_{40°C} = 34$), pupae ($n_{30°C} = 17$, $n_{40°C} = 34$), immature adults ($n_{30°C} = 17$, $n_{40°C} = 17$) and mature adults ($n_{30°C} = 17$, $n_{42°C} = 17$). The heat stress timings across development matched that for the previous life stage assays above. Following each heat stress application, males were held for 24 h in common garden conditions of 30°C then frozen at −20°C. Developmentally matched 30°C controls were frozen simultaneously with each heat stress treatment. The potential for testis recovery was assessed in the immature ($n_{40°C} = 17$) and mature ($n_{42°C} = 17$) adult treatments by freezing another cohort of males 25 days after heat exposure.

All males were dissected in saline buffer (0.9% NaCl solution) under a Zeiss Discovery V.12 stereomicroscope (Carl Zeiss, Jena, Germany) at ×20 magnification. The abdomen of each male was isolated from the thorax using fine-0.10*0.06 mm-tipped dissecting forceps (Dumont, Switzerland),

then the gastrointestinal and lower reproductive tracts were removed by separating them from the anal plates. The testes were gently excised from the softer dorsal side of the abdomen (under the elytra), moved to 30 µl of buffer on a cavity slide, and sealed under $20 \times 20$ mm coverslips. Images were viewed using an Olympus ×10, 0.25 NA Ach Ph1 objective on phase 2 of an Olympus BX41 microscope (Olympus Corporation, Tokyo, Japan), then captured with a GXCAM CCD camera (GT Vision, Suffolk, UK) and GXCapture (release 8.5) software. Fiji was used for image analysis [38]. *Tribolium castaneum* testes each contain six ellipsoid follicles (electronic supplementary material, figure S4). Each follicle's volume was calculated from its circumference. As follicles are fragile, sometimes they were damaged, consequently, the total testes volume for each male was calculated as the mean volume of its intact follicles, multiplied by 12. On average, $10.1 \pm 2.6$ s.d. intact follicles per male were measured, and there is significant covariation between follicle volumes within males ($S_{236} = 2092$, $p < 0.001$, $R_{s} = 0.86$, electronic supplementary material, figure S5).

### 2.6.2. Sperm counts and viability

Heat stress impacts on sperm production and its recovery were measured. Males were assigned to 30°C control ($n = 14$) or 42°C heat stress ($n = 16$) conditions, then paired with standard control virgin mature females at 1 and 25 days after the period of thermal exposure (electronic supplementary material, figure S3). To encourage successful inseminations, control males were provided with female access for 90 min, and heat-exposed males for 210 min (because heat-stressed males can take longer to mate; [23]). After mating, spermatophores were dissected from the female tracts, and sperm dispersed and diluted so that subsamples could be counted. Methods for measuring *T. castaneum* sperm number and viability have been previously detailed [34]. Briefly, sperm were released from spermatophores and dispersed evenly in Grace's insect buffer, subsamples of cells counted, and total sperm count calculated using the dilution factor. Sperm viability was measured using the LIVE/DEAD Sperm Viability Kit L-7011 (Molecular Probes, Oregon, USA) [39], which contains a SYBER-14 dye that stains all viable sperm heads with intact cell membranes green, and propidium iodide which enters the sperm heads of non-viable cells with compromised nuclear membranes and stains them red [40]. On average, $231.2 \pm 123.0$ s.d. sperm were screened per spermatophore to assay viability, and total sperm count was calculated for each spermatophore following six subsample counts.

## 2.7. Statistical analyses

Data were analysed using RStudio.0.99.903 [41] in R.3.3.2 [42]. Boxplots were created using 'ggplot{ggplot2}' [43], displaying the jittered raw data points, a mean dot, median line, interquartile range (IQR) boxes, and 1.5*IQR whiskers. All data were analysed using generalized linear models (GLMs) in 'glm{stats}' [42], or generalized linear mixed models (GLMMs) 'glmer{lme4}' [44] where undesired variation in the response variable caused by random factors needed to be accounted for [45,46]. After assessing the overall treatment significance, simple *post hoc* comparisons between treatment groups and controls were derived from 'summary{model}' [44]. 'lsmeans{lsmeans}' [47] was used when pairwise Tukey comparisons were necessary, and alpha values were corrected for false-positive results [46]. As a measure of how much variation in the response variable was explained by the model, pseudo $R^2$ (explained deviance) was calculated for GLMs ([48], p. 87). For GLMMs, 'r.squaredGLMM{MuMIn}' [49] was used to report the marginal $R^2$ as the variation explained by the fixed factors, and conditional $R^2$ explained by fixed and random factors.

   *Proportions of individuals surviving heat stress exposure* were analysed using GLMs with logit-linked quasi-binomial distributions. The response variable was entered as a paired 'cbind(success, fail){base}' variable, where 'success' was the number of heat stress survivors, and 'fail' was the number of deaths [48]. In the quasi-binomial GLM analysing the post-heat stress survival of males and females, the sex (male or female), the heat stress treatment (control or heat) and their interaction were entered as fixed factors. The life stage-specific heat stress survival assay was not fully factorial, therefore, its quasi-binomial GLM analysis contained a single fixed factor of 'life stage' (40°C, 42°C larvae; 40°C, 42°C pupae; 40°C, 42°C immature adults; 42°C mature adults; and 30°C controls). There was no variation in survival within 42°C-exposed pupae as all perished, causing complete separation in the model and preventing reliable parameter estimates [50]. Consequently, 42°C-exposed pupae were omitted from the main GLM, and differences between this treatment and other life stages made using one-sample Wilcoxon signed-rank tests. Tests were run with 'wilcox.test(mu=$x$){stats}', where the mean was set to zero to represent the pupal complete mortality.

The *sex-specific effect of heat stress on reproductive fitness of mature adult males and females* was analysed using a log-linked negative-binomial GLMM, with heat stress treatment entered as a fixed factor (controls; heated males with control females; both sexes heated), and the experimental repeat (1–2) as a two-level random factor. Excessive zero-inflation, with 25% of cases producing no offspring, prevented model convergence. Therefore, a zero-inflated negative-binomial hurdle GLMM was fitted [48], which consisted of a zero-presence model and a count model on the non-zero subset of the data. The frequency of zero cases across treatments was fitted with a logit-linked Bernoulli binomial GLMM, where zeros were coded as zero, and non-zeros as one. The non-zero count data were analysed with a log-linked negative-binomial GLMM.

The *life stage-specific effect of heat stress on subsequent reproductive fitness of males* (as mature adults) was analysed with a log-linked negative-binomial GLM, with the exposed life stage (larvae, immature adults, mature adults and the control) entered as a fixed factor. Pupae were omitted from analysis as none had survived.

For the *reproductive fitness recovery assays*, the same male individuals were re-mated to new females at successive time-points. Therefore, to prevent temporal autocorrelation, each assay was analysed as a log-linked Poisson GLMM with repeated measures, which included random intercepts for re-mating time-points and random slopes for individual male identity ([46], p. 642). The recovery assay GLMM exploring mature male reproductive fitness following single- or double-heat stress periods, contained male treatment (control, one or two heat stresses), the interval between heating and mating (5, 10, 15 or 20 days) and their interaction as fixed effects. The assay tracking the post-heat stress reproductive fitness recovery of different life stages, was analysed with a GLMM including male treatment (heated immature adults males or mature males, and controls), the heat stress to mating interval (1, 6 or 23–28 days), and their interaction, as fixed effects. However, the maximal GLMM could not converge with imbalanced data, as the mature male treatment had no representation at the day 6 time-point. Consequently, the asynchronous time-point factor was split into a log-linked negative-binomial GLM analysing day 6 between control and immature adult treatments, and a repeated measures log-linked Poisson GLMM analysing day 1 and days 23–28 to compare between all treatments [46].

For *all assays of testes volumes* (in ml³), the male heat stress treatment was fitted as the fixed factor. Elytra length (in mm), representative of body size, was influential for testes volumes and was affected by heat treatment in all life stages besides mature adults (electronic supplementary material, figure S6). Consequently, elytra length was included as a random factor for the larval, pupal and recovery testes assays. All three datasets generated relatively normally distributed continuous data, so were analysed with identity-linked Gaussian GLMMs. The successive heat stress assay produced skewed continuous data, which were fitted with a log-linked Gamma GLM.

The assay tracking the recovery of *sperm number and viability* sampled the same males at successive time-points. Therefore, repeated measures GLMMs were used, which included random intercepts for re-mating time-points and random slopes for individual male identity ([46], p. 695). For both sperm recovery assays, the heat stress treatment, the heat stress to mating interval (1 or 25 days), and their interaction, were included as fixed effects. Sperm count recovery was analysed with a log-linked Poisson GLMM. Sperm viability recovery was analysed using a logit-link binomial GLMM, where the response was a paired 'cbind(success, fail){base}' variable with 'success' being the live green sperm head frequency and, 'fail' the dead red head frequency.

# 3. Results

Simplified model summaries, including analysis of deviance tests on factors of interest, degrees of freedom and *p*-values, are reported in electronic supplementary material, table S1. Electronic supplementary material, table S2 presents detailed information, including model structures, factor-level betas and *t*/*z*-statistics for the relevant cross-treatment individual comparisons.

## 3.1. Sex and life stage sensitivities

### 3.1.1. Mature adult life stage

Exposure to 5-day periods of 42°C heat stress reduced mature adult survival ($\chi^2_{1,133} = 209.9$, $p < 0.001$; figure 1). There was no sex-specific difference in survival between groups in benign 30°C control conditions (95 ± 1%) ($z = -0.7$, $p = 0.885$, $n_{male} = 30$, $n_{female} = 39$), but males survived heated conditions

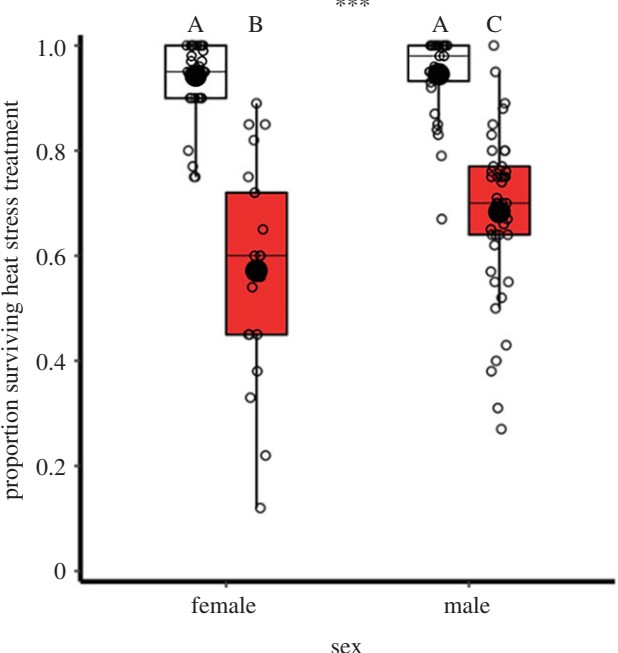

**Figure 1.** Survival rates of mature adults following exposure to heat stress conditions. Comparison of sex-specific survival through 42°C 5-day heat stress (red) relative to controls at 30°C (white). Samples sizes, left to right: $n = 39, 21, 30, 45$, where each data-point was a group (containing $43.2 \pm 47.8$ s.d. individuals). Significance threshold: $^{***} = p < 0.001$.

better than females ($\chi^2_{1,133} = 9.1$, $p = 0.003$, $n_{male} = 45$, $n_{female} = 21$). Relative to sex-matched controls, heat stress reduced male survival by 28% ($z = 8.3$, $p < 0.001$), and females by 39% ($z = 9.9$, $p < 0.001$).

   As in previous work [23], mature male reproductive fitness (measured as offspring production after 2 days of mating and 10 days of female oviposition), was significantly impacted by previous exposure to heat stress conditions ($\chi^2_{2,204} = 14.6$, $p < 0.001$; figure 2 and electronic supplementary material, table S1); males showed a halving in the number of offspring sired after being exposed to 5-day 42°C thermal conditions, compared with 30°C controls ($z = -2.4$, $p < 0.001$, $n_{30°C} = 134$, $n_{42°C} = 114$). When both mature males and females were exposed to heat stress, there was a similar halving in reproductive output ($z = -3.9$, $p < 0.001$, $n_{both} = 33$), but no additive effect from the combined exposure of females ($z = -2.1$, $p = 0.101$). Male heat stress exposure did not affect offspring sex ratio ($V_{(17)} = 79$, $p = 0.587$, $n = 17$; electronic supplementary material, figure S7).

### 3.1.2. Immature and juvenile life stages

Sensitivity to surviving 5-day heat stress conditions was specific to the temperature and life stage exposed ($\chi^2_{6,175} = 742.8$, $p < 0.001$; figure 3 and electronic supplementary material, table S1). The only treatment with group survival comparable to controls was the 40°C larval heat exposure ($z = 1.6$, $p = 0.108$, $n_{control} = 68$, $n_{larvae} = 9$), while 40°C conditions caused a 33% reduction in pupal survival ($z = -7.6$, $p < 0.001$, $n_{pupae} = 17$) and 50% for immature adults ($z = -9.4$, $p < 0.001$, $n_{immature} = 10$). Heat stress periods at 42°C were more lethal. Mature adults proved the most resistant stage, with a 33% reduction in survival following 42°C exposure relative to controls ($z = -8.0$, $p < 0.001$, $n_{mature} = 18$). Immature adults were the second most resistant stage following 42°C conditions, with an 80% decrease in survival ($z = 14.2$, $p < 0.001$, $n_{immature} = 22$), followed by larvae showing a 90% loss ($z = 17.2$, $p < 0.001$, $n_{larvae} = 38$). Pupae were the most sensitive life stage, with complete mortality following 42°C heat stress ($n_{pupae} = 5$).

   The life stage at which thermal stress exposure occurred had significant consequences for subsequent reproductive fitness at the mature adult male stage ($\chi^2_{3,104} = 143.4$, $p < 0.001$; figure 4 and electronic supplementary material, table S1). Males exposed to heat stress conditions as late instar larvae showed no decline in reproductive fitness as adults, compared with control males ($z = -0.1$, $p = 0.999$, $n_{control} = 29$, $n_{larvae} = 19$). However, immature males exposed to heat stress through the first 5 days of their adult life were exceptionally vulnerable, showing almost complete reproductive sterility at subsequent reproductive maturity (electronic supplementary material, table S1; comparison with controls: $z = -13.9$,

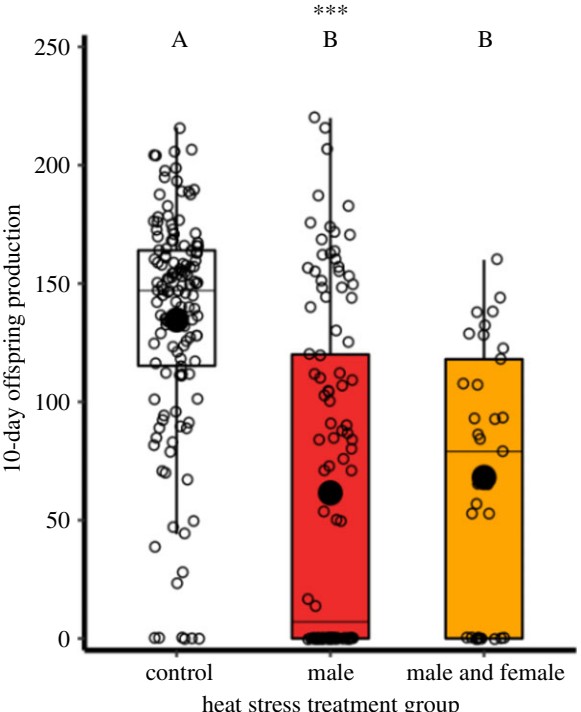

**Figure 2.** Impact of heat stress on reproductive fitness. Sum of 10-day reproductive fitness of control males ($n = 134$) compared with heat-exposed males ($n = 114$), and a treatment where both sexes were previously heat-exposed ($n = 33$). Significance threshold: $^{***} = p < 0.001$.

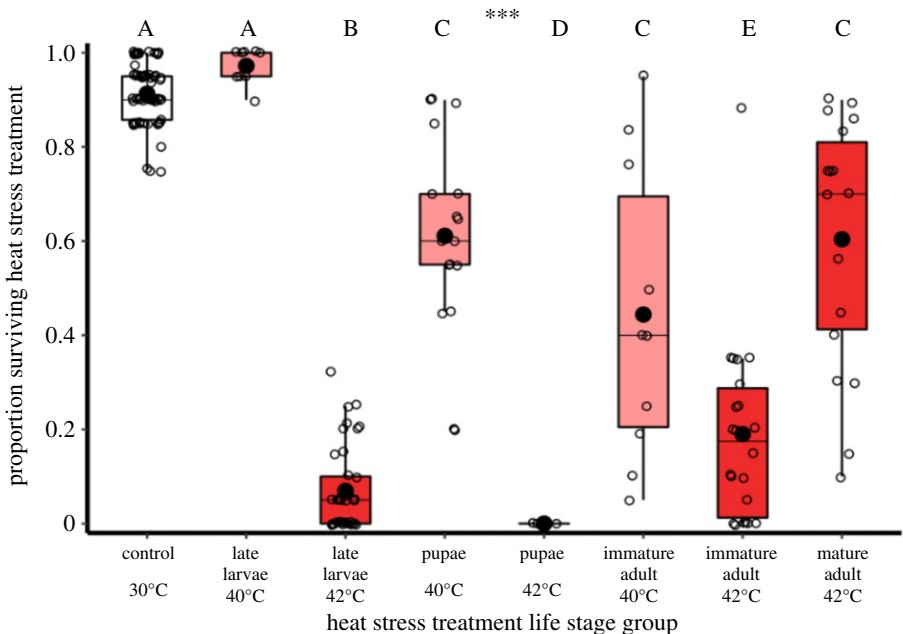

**Figure 3.** Life stage survival rates following exposure to heat stress. Sample sizes, left to right: $n = 68, 9, 38, 17, 5, 10, 20, 18$. Data points are groups (containing $19.5 \pm 4.5$ s.d. individuals). Differences to $0 \pm 0$ survival of 42°C-exposed pupae were analysed using one-sampled Wilcoxon rank tests. Significance threshold: $^{***} = p < 0.001$.

$p < 0.001$; heat-exposed larvae: $z = -12.7$, $p < 0.001$; heat-exposed mature adults: $z = -11.6$, $p < 0.001$, $n_{\text{immature}} = 30$, $n_{\text{mature}} = 30$). As previously shown, heat stress exposure for mature adults caused significant declines in male reproductive fitness relative to controls ($z = -2.8$, $p = 0.027$; figures 5, 6a,b). Since all pupae perished through heat stress, no subsequent reproduction was measured for this life stage.

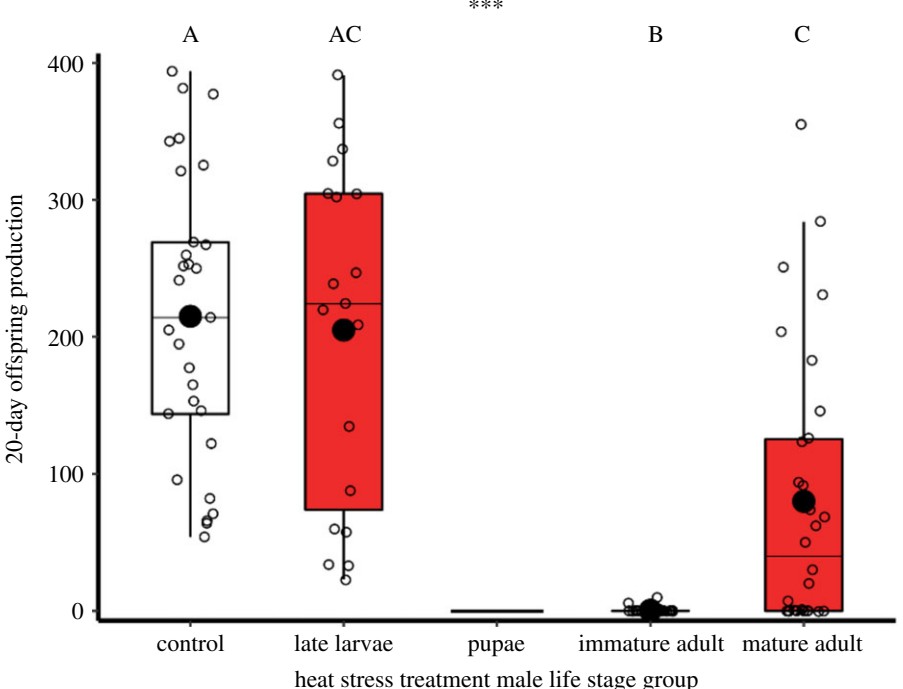

**Figure 4.** Impact of 42℃ heat stress conditions at different life stages on subsequent reproductive fitness of mature adult males. Line at zero for the pupal treatment represents zero reproductive fitness due to complete mortality during heat exposure. Sample sizes left to right: $n = 29, 19, 30, 30$. Significance threshold: *** = $p < 0.001$.

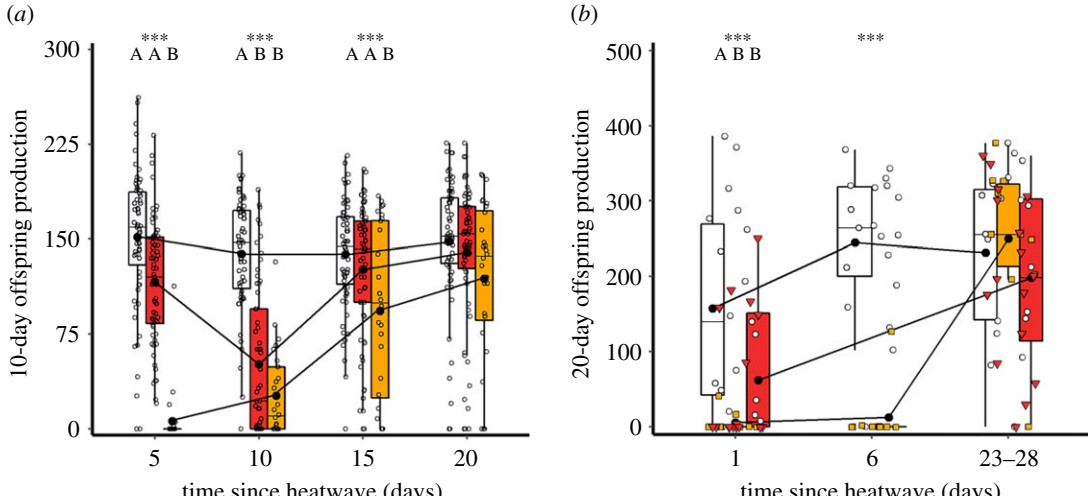

**Figure 5.** Recovery of male reproductive fitness following exposure to single- or double-heat stress conditions (*a*), or immature versus mature adult male life stages. (*a*) Male reproductive fitness following exposure to 30℃ control (white, $n = 63$), single 42℃ heat stress conditions (red, $n = 63$) and double 42℃ heat stress conditions (orange, $n = 28$). (*b*) Male reproductive fitness following exposure to 30℃ control (white circles, $n = 19$), a 42℃ period of heat stress during the immature adult stage (orange squares, $n = 10$), and 42℃ heat stress exposure to the mature adult life stage (red triangles, $n = 16$). Mature males were not assayed in the day 1 time-point as, being from the same cohort, they were then undergoing heat stress treatment. Within time-point significance threshold: *** = $p < 0.001$.

## 3.2. Recovery of male reproductive fitness

### 3.2.1. Recovery of mature males following exposure to single- and double-heat stress conditions

Overall, when male reproductive fitness following heat stress exposures was measured over 20 days across four mating opportunities (males provided with new females every 5 days), males exposed to a

single-heat stress periods sired 25% fewer offspring compared with controls, while the total reproductive fitness of males exposed to two heat stress periods was reduced by 58% ($\chi^2_{2,151} = 94.0$, $p < 0.001$; figure 5a, $n_{30°C} = 63$, $n_{42°C} = 63$, $n_{2*42°C} = 28$). Generally, the mean reproductive fitness of males changed significantly over time ($\chi^2_{3,150} = 59.6$, $p < 0.001$), with different recovery trajectories between treatments reflected by a significant interaction between heat exposure and recovery time ($\chi^2_{6,146} = 190.4$, $p < 0.001$). Reproductive fitness of control males was consistent across the four mating opportunity assays on days 5, 10, 15 and 20 (figure 5a, electronic supplementary material, table S2). However, heat-treated males showed significant variation in reproductive output over time. Offspring production following mating opportunities on day 10 was lower than on day 20, for both single- ($z = -9.0$, $p < 0.001$) and double-heat stress treatments ($z = -6.6$, $p < 0.001$), but not for controls ($z = -0.3$, $p = 1.000$). Compared with control males, exposure to one period of heat stress caused a significant 63% decline in reproductive fitness for the first 10 days following the treatment ($z = -6.4$, $p < 0.001$). Afterwards, males showed reproductive recovery, with productivity becoming similar to controls by day 15 ($z = -1.2$, $p = 0.991$) and 20 ($z = -0.4$, $p = 1.000$; figure 5a).

Exposure to two periods of heat stress caused mature male reproductive output to decline by 97% (compared with controls), when provided with mating opportunities on the fifth day after the end of heat treatment ($z = -16.7$, $p < 0.001$). Males exposed to two heat stress conditions also had a lower reproductive fitness than single-heat stress males following mating opportunities at day 5 ($z = -15.9$, $p < 0.001$). By day 10, reproductive fitness of double-heat stress-exposed males remained minimal compared with controls ($z = -6.4$, $p < 0.001$); however, it was now equivalent to single-heat stress-exposed males, which had worsened ($z = -1.5$, $p = 0.923$). Reproductive recovery of the double-heat stress-exposed males became evident after 15 days, though at a depressed rate as offspring productivity was 67% of controls ($z = -4.3$, $p < 0.001$) and 74% of the single-heat stress group ($z = -3.4$, $p = 0.038$). After 20 days following double-heat stress exposure, male reproductive fitness had recovered to that of control and single-heat stress males ($z = -2.1$, $p = 0.588$ and $z = -1.8$, $p = 0.796$; figure 5a).

### 3.2.2. Recovery of immature adult males from single periods of heat stress exposure

The overall comparison of reproductive recovery by immature males (exposure commenced within 1 day of adult eclosion) versus mature males (exposure commenced at 10 days following adult eclosion) and their respective controls following a single period of heat stress did not yield a significant difference between treatment groups ($\chi^2_{2,42} = 3.8$, $p = 0.159$; figure 5b). However, specific comparisons yielded major differences between treatments and time. Male reproductive fitness across treatments improved significantly between day 1 and 23–28 days after heat stress exposure ($\chi^2_{1,43} = 25.1$, $p < 0.001$, $n_{control} = 19$, $n_{immature} = 10$, $n_{mature} = 16$). A significant interaction across time revealed that recovery trajectories differed between the treatment groups ($\chi^2_{2,42} = 21.7$, $p < 0.001$). Immature males provided with a mating opportunity 1 day after the heat treatment achieved only 4% of offspring production compared with controls ($z = -4.6$, $p < 0.001$). This depression in offspring production by immature adult males following heat exposure continued for at least 6 days, when reproductive fitness was still only 5% that of control males ($\chi^2_{1,27} = 13.7$, $p < 0.001$; $z = -4.6$, $p < 0.001$). However, by 23–28 days since the increased thermal exposure, offspring production was comparable to control males ($z = 0.1$, $p = 1.000$). In comparisons between the recovery of adult males when exposed to heat stress at the immature or mature life stage, there was no difference in reproductive fitness 1 day after exposure ($z = 2.1$, $p = 0.293$), with both achieving one quarter that of control males ($z = -3.2$, $p = 0.016$). After 23–28 days recovery from heat stress exposure, offspring output the reproductive fitness of mature males was similar to immature males ($z = -0.5$, $p = 0.998$) and controls ($z = -0.6$, $p = 0.990$; figure 5b).

### 3.3. Testis size and spermatozoal recovery assays

Heat stress exposure caused significant decreases in testis volume among mature adult males ($F_{2,46} = 60.3$, $p < 0.001$; figure 7a). A single 5-day 42°C period of heat stress reduced testes volumes by 39% compared with control males ($t = -3.7$, $p < 0.001$, $n_{30°C} = 16$, $n_{42°C} = 16$). Two sequential heat stress periods (with a 10-day interim under standard 30°C conditions) decreased testes volumes by 71% compared with controls ($t = -9.7$, $p < 0.001$, $n_{2*42°C} = 17$) and by 52% compared with males exposed to a single period of heat stress exposure ($t = -5.9$, $p < 0.001$; figure 7).

Heat stress exposure at 40°C during the larval stage had no impact on testis volumes as mature adults ($F_{1,49} = 1.9$, $p = 0.169$; $t = -1.6$, $p = 0.100$, $n_{30°C} = 16$, $n_{42°C} = 34$; figure 7b). Contrastingly, 40°C heat

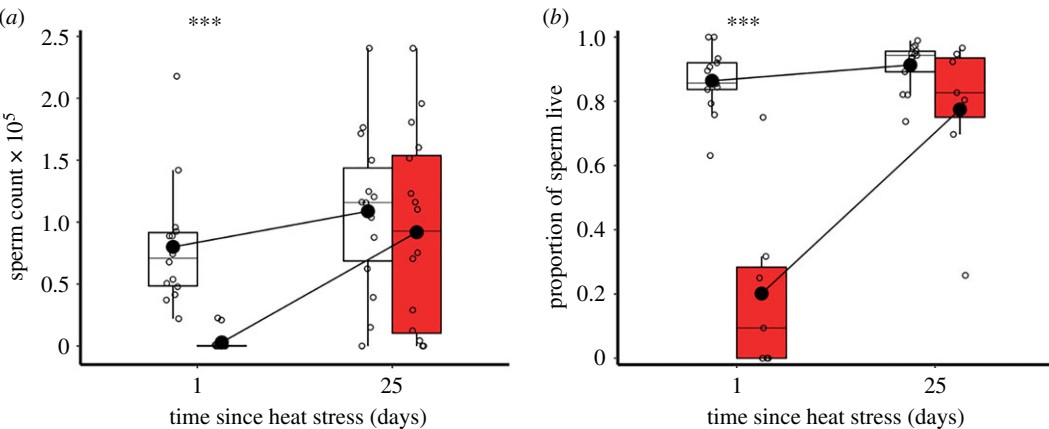

**Figure 6.** Recovery of sperm counts and viability after exposure to heat stress. (*a*) Mean spermatophore sperm counts of mature males following exposure to 30°C control (white bars, $n = 14$) and 42°C heat stress conditions (red bars, $n = 16$). (*b*) Proportion of viable sperm in mature males following exposure to control (white bars, $n = 13$) and heat stress (red bars, $n = 7$) conditions. Sperm counts and viability were derived from the sum of six technical replicates per individual male. Significance threshold: $*** = p < 0.001$.

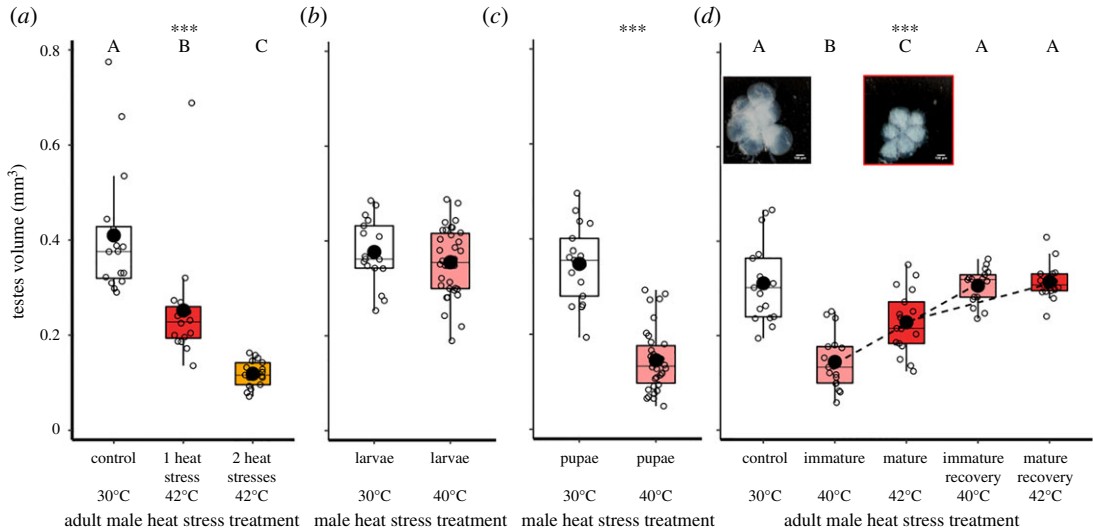

**Figure 7.** Impacts of previous heat stress conditions on the development of mean testes volumes of adult males. (*a*) Males exposed to control ($n = 16$), single- ($n = 16$), or double-heat stress ($n = 17$) conditions. (*b*) Males treated with control ($n = 16$) or heat stress ($n = 34$) conditions as larvae. (*c*) Males exposed to control ($n = 16$) or heat stress ($n = 34$) conditions as pupae. (*d*) Testes volumes of controls, and males exposed to heat stress as freshly eclosed immature, or 10($\pm$2)-day old mature ($n = 34$) adults. Sizes were measured 1 and 26 days following heat stress treatment, standardized to the mature adult treatment. Dotted lines represent linked recovery treatments, which are not truly repeated measures. Sample sizes left to right: $n = 17, 17, 17, 17, 17$. Significance threshold: $*** = p < 0.001$. Inset, dark-field phase-contrast images of testes of mature adults from control (white) and heat stress (red) treatments.

stress exposure to the pupal stage reduced mature adult testis sizes by 61% compared with controls ($F_{1,49} = 53.2$, $p < 0.001$; $t = -9.5$, $p < 0.001$; figure 7*c*, $n_{30°C} = 16$, $n_{42°C} = 34$). Moreover, males exposed to heat stress as immature adults showed a relative reduction of 63% on maturity ($F_{4,80} = 71.1$, $p < 0.001$; $z = -4.0$, $p < 0.001$, $n_{control} = 17$, $n_{immature} = 17$, $n_{mature} = 17$; figure 7*d*).

Males could recover from heat stress damage to testes, with the volumes of 42°C heat stress-treated mature ($t = -0.3$, $p = 0.999$) and 40°C heat stress-treated immature adults ($t = 0.2$, $p = 0.999$) returning to control levels 25 days after exposure.

After mature adult males were exposed to 5-day 42°C heat stress conditions, overall sperm counts declined by 51% compared with controls ($\chi^2_{1,28} = 7.6$, $p = 0.006$, $n_{30°C} = 14$, $n_{42°C} = 16$; figure 6*a*). Specifically, sperm counts of heat stress-exposed males were 91% lower than that of controls on day 1 ($z = -5.1$, $p < 0.001$), but by day 25 they had recovered to be comparable ($z = -1.1$, $p = 0.716$). Sperm

counts for both treatments increased over 25 days ($\chi^2_{1,28} = 6.1$, $p = 0.013$), but a significant interaction highlighted a much faster trajectory for heat-exposed males ($\chi^2_{1,28} = 12.8$, $p < 0.001$). Specifically, control male sperm counts remained consistent between days 1 and 25 ($z = 0.5$, $p = 0.969$), while heat-treated males showed a significant increase over the 25-day recovery time ($z = 5.1$, $p = 0.002$).

The overall sperm viability model showed no combined statistical differences between treatment ($\chi^2_{1,18} = 2.3$, $p = 0.130$, $n_{30°C} = 13$, $n_{42°C} = 7$) or time-points ($\chi^2_{1,18} = 1.5$, $p = 0.214$; figure 6b). However, there was a significant interaction between heat stress exposure and time, with the recovery profiles for sperm viability differing between control and heat stress treatments ($\chi^2_{1,18} = 10.0$, $p = 0.002$). In particular, sperm viability of heat stress-exposed males was 77% lower than controls on day 1 ($z = -3.7$, $p < 0.001$). The sperm viability of control males remained consistent over the 25-day recovery period ($z = 0.7$, $p = 0.893$), while the sperm viability of heat stress-treated males improved significantly between day 1 and 25 after exposure ($z = 3.6$, $p = 0.002$), and was similar to controls by day 25 ($z = -2.0$, $p = 0.176$).

## 4. Discussion

Our experiments exposed *T. castaneum* beetles to stable 40°C or 42°C 5-day heat stress conditions that were 5–7°C above the optimum for population productivity in *T. castaneum* [26,27]. Following this exposure, we find that: (i) heat stress conditions reduce survival and damage male reproduction, (ii) different life stages show varying sensitivity to heat stress, and (iii) recovery from damage to reproductive function can occur between 15 and 28 days following the end of exposure to thermal stress.

Overall, a single 42°C period of heat stress killed all pupae, approximately 40% of mature adults, 80% of immature adult males and 90% of larvae (figure 1). Weaker effects were found for 40°C heat stress conditions, and equivalent mortality among mature control adults not exposed to heat stress was approximately 10%. These findings identify wide variation in heat stress survival across life stages in this insect system. Previous work, where shorter-term experimental 'heat shocks' were applied to *T. castaneum* also identified lower survival in earlier life stages [51], including developing eggs [27]. In the context of climate change impacts on natural insect populations, the extreme thermal sensitivity of pupae, with all individuals perishing, is particularly concerning for biodiversity. Higher heat-induced mortality during intermediate life stages such as larvae and pupae has been reported in other insects [52,53]. Several hypotheses exist for the vulnerability of these stages including thinner and more permeable cuticles [54], limited dispersal capabilities [24] and disruption to complex hormone and gene expression associated with development [55]. Observational UK butterfly climate response data also suggest that the pupal life stage has greater sensitivity to extreme events [56]. Considering that most insect species are holometabolous endopterygotes [57], pupal sensitivity is a particular area for further research.

Heat stress damaged reproductive function, and different life stages showed varying reproductive vulnerability to heat exposure. Although heat stress-exposed late instar larvae showed no decrease in testis size or reproductive fitness when they emerged into mature adult males, immature adult males exposed to a period of heat stress suffered almost complete reproductive sterility as mature adults, and sired very few offspring. These impacts reveal a heightened vulnerability for freshly eclosed males in the process of maturing into the adult reproductive stage which, in *T. castaneum*, takes at least 10 days [58]. Likewise, we found extreme sensitivities during the pupal stage to thermal stress, with complete mortality under 42°C heat stress simulations. A milder 40°C period of heat stress in the pupal stage revealed reproductive damage, with mature adult testes volumes having decreased by two-thirds (figure 7c). Sexually mature adult males lost about half their reproductive potential with one 5-day 42°C period of heat stress (figures 2 and 4), as previously reported [23]. The combined exposure of both male and female mature adults to thermal stress had no additive or antagonistic effects (figure 2), suggesting male-specific reproductive vulnerability to hyperthermia, as seen in other ectotherms [20–22,24,59–61]. Exposing mature adults to two periods of heat stress, with a 10-day interim period at 30°C, strengthened the reproductive impact further, with males showing near-sterility and slow recovery (figure 5a). Comparisons between double- and single-heat stress effects on male reproduction, therefore, revealed an additive effect of elevated thermal challenge, and no evidence for 'hardening' [62].

Assays of testis size, sperm number and sperm viability revealed parallel declines in these key traits at different life stages alongside male reproductive fitness. There was a significant reduction in the volume of male gonadal tissue after heat stress exposure (figure 7), and rapid declines in ejaculate sperm number and viability (figure 6). In recent work, we also observed failure by these damaged sperm to access the

fertilization set within female reproductive tract storage sites [23]. A number of ectotherm studies show that sperm counts can be impacted by thermal stress [59,60,63–66]. Increased temperatures have also been shown to affect testis size [67], sperm viability [68,69], head morphology [63], sperm tail length and nuclear integrity [65].

Several hypotheses have been proposed for why sperm and/or spermatogenesis may be especially vulnerable to thermal stress, compared with ova and oogenesis. Spermatogenesis is a continuous process with many more divisions which present opportunities for divisional or mutational errors [70]. Mature sperm are highly specialized cells [71] with potentially heat-sensitive properties such as flagellar motility and condensed nuclear material [14]. Moreover, relative to eggs, sperm have a large surface-area-to-volume ratio, possibly reducing the heat-buffering capability of the cytoplasm [72]. Compared with ovaries and embryonic tissue, testis repair systems have lower levels of HSP expression [73], and incomplete DNA repair pathways [74], allowing heat to denature proteins integral for metabolism and structural or genetic integrity [75]. Sperm count could also decline due to cell lysis from controlled apoptosis or necrosis in the testis [76], consistent with a decline in testis size, sperm number and previous observations of cell degradation in ejaculates from heat-exposed males [23]. Heat-exposed sperm may also suffer DNA breakage [77], and chromatin decondensation [13,15], leading to blastocyst and embryo failures [78]. Thermal damage to the primary nucleotide structure, epigenetic processes (such as methylation or histone modifications) or seminal fluid proteins, could also disrupt embryogenesis [16,71]. The effect of 28°C heatwave conditions on a range of male fertility parameters in the ectothermic mussel (*Mytilus galloprovincialis*) showed successional and combined damage to multiple sperm traits through time from sperm counts to sperm motility and finally DNA fragmentation and abnormality [63].

Despite the initial damage to fertility, reproductive recovery was clearly possible for male *T. castaneum* beetles. Mature adult males showed reproductive recovery 15 days after exposure to one period of heat stress, but required 20 days following two such stress periods (figure 5*a*). Although heat exposure to reproductively immature males led to their complete sterility as mature adults, recovery was also possible, but over a longer 28-day period (figure 5*b*). In tandem with these assays of male reproductive fitness, testes volumes, sperm counts and sperm viability also recovered to control levels when measured 25 days after the end of the thermal stress treatment (figures 6 and 7). These patterns of recovery within 15–28 days are reassuring findings for longer-lived insects like *T. castaneum* [23], especially for the immature adult stage which seems so vulnerable to thermal stress.

The eventual post-heat stress reproductive recovery in *T. castaneum* (figures 5 and 6) and other ectotherms [65,79], and the resistance of earlier life stages to reproductive damage (figure 4) [80], suggest that it is later stages of sperm production which are more sensitive to thermal stress. In recent work, we demonstrated that mature sperm stored within the female are also vulnerable to heat exposure, leading to a halving of subsequent fertility, and transgenerational impacts on offspring reproduction and lifespan [23]. If later sperm production stages are more sensitive to thermal stress, it is possible that new and undamaged spermatogenic cycles can begin once the period of heat exposure has passed, allowing recovery through time. In experiments on endotherms, fertility recovers over one to two months after thermal stress [15,16]. Ectotherms have received far less investigation here, but *Drosophila melanogaster* adult males require approximately 10 days to recover from sterility after 29°C thermal stress, coinciding with the length of a spermatogenesis cycle [65,66]. Recovery time is thought to be associated with the total length of spermatogenesis [10,65], explaining why recovery is slower in endotherms where spermatogenesis can take weeks or months, rather than the days or weeks in ectotherms [13,14,81]. Initial production of spermatocytes and early spermatids generally begins in the larval stages in endopterygote insects [27,82]. Our experiments reveal vulnerability from the pupal stage, when the male reproductive system is active with spermiogenesis and the formation of spermatids with tails, mitochondria and nuclear condensation [83–85].

What is the relevance of our study for insects exposed to heatwaves in nature? We have taken an experimental approach to advance our initial understanding of the impacts of thermal stress associated with heatwave conditions, isolating stable heat stress as a causative factor for reproduction and survival, and using one representative system. In the natural environment, the option to move and/or seek microclimatic relief from thermal stress will be options for mobile animals, so research into the impacts of natural heatwaves on insects are obviously a priority. Moreover, natural thermal environments are rarely completely stable, so we plan further measures of heatwave impacts where experimental temperatures are varied and the sum of heat exposure and its variability can be assessed [86]. Despite natural variation and our results showing an ability for recovery, our combined survival

and reproductive fitness data under stable thermal maxima highlight detrimental impacts of experimental heatwave conditions on *T. castaneum* that persist for two to four weeks, especially affecting male reproductive traits and in juvenile life stages. There is mounting evidence for the serious decline of insects in the wild, even in areas seemingly shielded from habitat change or pesticides, such as German nature reserves [30] and protected forests in Puerto Rico [31]. Although we cannot conclude that our experimental findings explain such declines in insect abundance and diversity in the natural environment, reduced survival and reproduction will obviously constrain a population's ability to persist or colonize, especially if faced with other stressors such as habitat loss. Two major recent studies of abundance and diversity declines in tropical arthropods and temperate bumblebees concluded that the most likely drivers of population collapses were increases in thermal maxima or unusually hot days [7,31].

Ethics. This study was approved by, and followed strict guidelines to, the University of East Anglia's Animal Welfare and Ethical Review Board.

Data accessibility. Datasets are available from Dryad Digital Repository at https://doi.org/10.5061/dryad.gmsbcc2kq [87].

Authors' contributions. M.J.G.G. and K.S. conceived and designed the experiments; K.S. and R.V. collected empirical data. K.S. conducted statistical analyses. K.S. and M.J.G.G. wrote the manuscript with input from R.V.

Competing interests. We declare we have no competing interests

Funding. This work was supported by a Natural Environment Research Council (NERC) project grant (NE/K013041/1) and NERC EnvEast DTP studentship (Award Ref: 1540234).

Acknowledgements. We thank Lewis Spurgin and Jo Godwin for advice on statistical analysis, Aldina Franco for useful comments on interpretation and Jessie Gardner for comments on the manuscript.

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
