## [Peer Review File · Royal Society Open Science]

Review History

RSOS-201717.R0 (Original submission)

Review form: Reviewer 1

Is the manuscript scientifically sound in its present form?

Yes

Are the interpretations and conclusions justified by the results?

No

Is the language acceptable?

Yes

Do you have any ethical concerns with this paper?

No

Have you any concerns about statistical analyses in this paper?

No

Recommendation?

Major revision is needed (please make suggestions in comments)

Comments to the Author(s)

This study investigates on the impacts of exposure to high temperature on the biology of different life stage in *Tribolium*: survival, offspring production, testis size and spermatozoal recovery. One interesting aspects is about the time necessary for reproduction rate to recover following exposure to high temperature depending on the life stage that experienced the heat treatment. In general, the paper is well written, the methods are very clear and easily reproducible from the information provided.

The main problem I have is that the study applied treatments with constant temperature, at either 40 or 42°C, for 5 consecutive days. In the field, during heat waves, the temperature is never constant at so high temperature for so long (daily fluctuations still operate). Therefore, the study cannot conclude on the impacts of heatwave conditions. Instead, this study provides effects of high temperature on the biology of this species (which is still of interest). But this is thermal biology study and not anymore a study on the effect of heatwaves. The word heatwave should be suppress almost everywhere in the main text (the same error can be found in previous studies published by same and other authors).

In addition, the control treatment consisted in an exposure to 30°C (constant, again). However, the optimal temperature is said to be around 35°C. Therefore, I wonder why the control was not fixed at 35°C ? That would increase the deviation between control and high temperature treatments, reflecting the extent to which a moderate increase in temperature is sufficient to generate high impacts (although this is not new). Indeed, at some point lower on the thermal performance curve, there should be a lower temperature at which the biological traits will show similar values than during the high temperature treatments, and one can wonder to what extent the 30°C is near this temperature equivalence.

The section on the statistical methodology is unusually long (4 pages) and should certainly be written in a more concise way by summarizing the first part on the general approach (the specific paragraphs after that provide the necessary details).

Review form: Reviewer 2

Is the manuscript scientifically sound in its present form?

Yes

Are the interpretations and conclusions justified by the results?

Yes

Is the language acceptable?

Yes

Do you have any ethical concerns with this paper?

No

Have you any concerns about statistical analyses in this paper?

No

Recommendation?

Accept with minor revision (please list in comments)

Comments to the Author(s)

I have read this paper a couple of times through and find the writing, analyses and interpretation are all clear. The work is well-justified, comprehensive and makes an excellent contribution to the growing literature, partially by this group, on better quantifying the potential role of climate-induced heat stress on insect reproduction. My only minor criticism is that the Discussion sometimes felt more like a review and if there are space considerations then I would recommend cutting down on line 779-852. The information in those sections are relevant but go somewhat beyond interpretation of the current results and future directions.

Review form: Reviewer 3

Is the manuscript scientifically sound in its present form?

Yes

Are the interpretations and conclusions justified by the results?

Yes

Is the language acceptable?

Yes

Do you have any ethical concerns with this paper?

No

Have you any concerns about statistical analyses in this paper?

No

Recommendation?

Accept with minor revision (please list in comments)

Comments to the Author(s)

This is a beautifully written manuscript on the consequences of heatwaves on the mortality of different life stages and, among the survivors, on the downstream effects on reproductive traits and function, using red flour beetles as a model organism. The authors found severe effects of heat stress on post-larval stages, with virtually all pupae dying or, in early imagoes, becoming sterile (including disrupted testicular development and reduced sperm viability) – at least temporarily.

This paper describes a carefully designed and executed experiment testing well-motivated hypotheses on a timely and relevant topic. It must be the most clearly and eloquently written manuscript of the many I have already peer-reviewed this year and probably one of my first to accept almost as is. My comments, therefore, are mostly of editorial nature. Chapeau to the authors!

Here a few minor comments for further clarification:

Lines 77-78: What do you mean by “increases in hotter environmental temperatures”? Simply increases in environmental temperature, increases in temperature particularly in hot climates (i.e. no change in more polar regions), or increases in average temperatures particularly of the hot season (i.e. no change in winter)?

Line 270: Is this error around mean post-eclosion time the SD or SE? Please define at first mention. And sample size = 49?

Line 505 and throughout: It would be helpful to remind the reader of the sample sizes (or degrees of freedom) associated with all these z-statistics as it seems cumbersome to go find them in the Methods or supplementary material.

Lines 505-507: Is this really such a coincidence that 3 p-values in a row are identical to three decimal places ($p = 0.671$) despite vastly different z-values, or were these results copy-pasted and accidentally not updated?

Line 601: What do you mean by “immature, mature males and controls”? Immature and mature males relative to control males or immature, mature and control males with one another? Or what are immature, mature males if not two different sets of males?

Line 863: sable -> stable

Decision letter (RSOS-201717.R0)

Dear Professor Gage

The Editors assigned to your paper RSOS-201717 "Fertility and mortality impacts of experimental heatwave conditions on different life stages and their reproductive recovery in a model insect" have now received comments from reviewers and would like you to revise the paper in accordance with the reviewer comments and any comments from the Editors. Please note this decision does not guarantee eventual acceptance.

Please submit your revised manuscript and required files (see below) no later than 21 days from today's (ie 23-Nov-2020) date. Note: the ScholarOne system will 'lock' if submission of the revision is attempted 21 or more days after the deadline. If you do not think you will be able to meet this deadline please contact the editorial office immediately.

on behalf of Dr Kristina Sefc (Associate Editor) and Pete Smith (Subject Editor)
openscience@royalsociety.org

Associate Editor Comments to Author (Dr Kristina Sefc):

Dear authors,

Your manuscript has been seen by three reviewers, who applaud your experiment and offer some advice for further improvement of your manuscript. Reviewer 1 comments on an important point, namely the simulation of natural heat waves, which involve fluctuations rather than stable temperature. This is a matter of what you claim to demonstrate in the manuscript, and can be resolved by revising your interpretation. When shortening the methods, please be careful to still include all information. Reviewer 1 also suggests that a 'read me first' text file should be added to the supplementary data, as the terms and headers in the tables are not comprehensible. Reviewer 2 recommends cutting down the text of the discussion. Although there are no space limits, please keep in mind that readers will appreciate a concise text. Please attend to all of the reviewers' comments, and I look forward to seeing your revision.

Reviewer comments to Author:
Reviewer: 1

Comments to the Author(s)

This study investigates on the impacts of exposure to high temperature on the biology of different life stage in *Tribolium*: survival, offspring production, testis size and spermatozoal recovery. One interesting aspects is about the time necessary for reproduction rate to recover following exposure to high temperature depending on the life stage that experienced the heat treatment. In general, the paper is well written, the methods are very clear and easily reproducible from the information provided.

The main problem I have is that the study applied treatments with constant temperature, at either 40 or 42°C, for 5 consecutive days. In the field, during heat waves, the temperature is never constant at so high temperature for so long (daily fluctuations still operate). Therefore, the study cannot conclude on the impacts of heatwave conditions. Instead, this study provides effects of high temperature on the biology of this species (which is still of interest). But this is thermal biology study and not anymore a study on the effect of heatwaves. The word heatwave should be suppress almost everywhere in the main text (the same error can be found in previous studies published by same and other authors).

In addition, the control treatment consisted in an exposure to 30°C (constant, again). However, the optimal temperature is said to be around 35°C. Therefore, I wonder why the control was not fixed at 35°C ? That would increase the deviation between control and high temperature treatments, reflecting the extent to which a moderate increase in temperature is sufficient to generate high impacts (although this is not new). Indeed, at some point lower on the thermal performance curve, there should be a lower temperature at which the biological traits will show

similar values than during the high temperature treatments, and one can wonder to what extent the 30°C is near this temperature equivalence.

The section on the statistical methodology is unusually long (4 pages) and should certainly be written in a more concise way by summarizing the first part on the general approach (the specific paragraphs after that provide the necessary details).

Reviewer: 2

Comments to the Author(s)

I have read this paper a couple of times through and find the writing, analyses and interpretation are all clear. The work is well-justified, comprehensive and makes an excellent contribution to the growing literature, partially by this group, on better quantifying the potential role of climate-induced heat stress on insect reproduction. My only minor criticism is that the Discussion sometimes felt more like a review and if there are space considerations then I would recommend cutting down on line 779-852. The information in those sections are relevant but go somewhat beyond interpretation of the current results and future directions.

Reviewer: 3

Comments to the Author(s)

This is a beautifully written manuscript on the consequences of heatwaves on the mortality of different life stages and, among the survivors, on the downstream effects on reproductive traits and function, using red flour beetles as a model organism. The authors found severe effects of heat stress on post-larval stages, with virtually all pupae dying or, in early imagoes, becoming sterile (including disrupted testicular development and reduced sperm viability) – at least temporarily.

This paper describes a carefully designed and executed experiment testing well-motivated hypotheses on a timely and relevant topic. It must be the most clearly and eloquently written manuscript of the many I have already peer-reviewed this year and probably one of my first to accept almost as is. My comments, therefore, are mostly of editorial nature. Chapeau to the authors!

Here a few minor comments for further clarification:

Lines 77-78: What do you mean by “increases in hotter environmental temperatures”? Simply increases in environmental temperature, increases in temperature particularly in hot climates (i.e. no change in more polar regions), or increases in average temperatures particularly of the hot season (i.e. no change in winter)?

Line 270: Is this error around mean post-eclosion time the SD or SE? Please define at first mention. And sample size = 49?

Line 505 and throughout: It would be helpful to remind the reader of the sample sizes (or degrees of freedom) associated with all these z-statistics as it seems cumbersome to go find them in the Methods or supplementary material.

Lines 505-507: Is this really such a coincidence that 3 p-values in a row are identical to three decimal places ($p = 0.671$) despite vastly different z-values, or were these results copy-pasted and accidentally not updated?

Line 601: What do you mean by “immature, mature males and controls”? Immature and mature males relative to control males or immature, mature and control males with one another? Or what are immature, mature males if not two different sets of males?

Line 863: sable -> stable

===PREPARING YOUR MANUSCRIPT===

===PREPARING YOUR REVISION IN SCHOLARONE===

<https://royalsociety.org/journals/authors/author-guidelines/#supplementary-material> to include a suitable title and informative caption. An example of appropriate titling and captioning may be found at https://figshare.com/articles/Table_S2_from_Is_there_a_trade-off_between_peak_performance_and_performance_breadth_across_temperatures_for_aerobic_sc_ope_in_teleost_fishes_/3843624.

Author's Response to Decision Letter for (RSOS-201717.R0)

See Appendix A.

RSOS-201717.R1 (Revision)

Review form: Reviewer 1

Is the manuscript scientifically sound in its present form?

Yes

Are the interpretations and conclusions justified by the results?

Yes

Is the language acceptable?

Yes

Do you have any ethical concerns with this paper?

No

Have you any concerns about statistical analyses in this paper?

No

Recommendation?

Accept as is

Comments to the Author(s)

All my comments have been considered with care - I thank the authors for that. I have absolutely nothing to add. This is an excellent paper. I still kind of disagree with the interpretation of the term heatwave, but the interpretation of the authors is quite clear such that the readers will have the opportunity to make their own idea. This will add to the hot debate about heatwave characteristics!

Review form: Reviewer 3

Is the manuscript scientifically sound in its present form?

Yes

Are the interpretations and conclusions justified by the results?

Yes

Is the language acceptable?

Yes

Do you have any ethical concerns with this paper?

No

Have you any concerns about statistical analyses in this paper?

No

Recommendation?

Accept as is

Comments to the Author(s)

The authors have addressed my comments adequately. I have no further comments except that the green stain of the LIVE/DEAD viability kit is called "SYBR", not "SYBER". Please correct this in the proofs.

Decision letter (RSOS-201717.R1)

Dear Professor Gage,

It is a pleasure to accept your manuscript entitled "Fertility and mortality impacts of thermal stress from experimental heatwaves on different life stages and their recovery in a model insect" in its current form for publication in Royal Society Open Science.

on behalf of Dr Kristina Sefc (Associate Editor) and Pete Smith (Subject Editor)
openscience@royalsociety.org

Associate Editor Comments to Author (Dr Kristina Sefc):

Associate Editor: 1

Comments to the Author:

Congratulations to the authors for their successful revision! Please remember to correct the name of the dye (SYBR) as noted by reviewer 2.

Reviewer comments to Author:

Reviewer: 1

Comments to the Author(s)

All my comments have been considered with care - I thank the authors for that. I have absolutely nothing to add. This is an excellent paper. I still kind of disagree with the interpretation of the term heatwave, but the interpretation of the authors is quite clear such that the readers will have the opportunity to make their own idea. This will add to the hot debate about heatwave characteristics!

Reviewer: 3

Comments to the Author(s)

The authors have addressed my comments adequately. I have no further comments except that the green stain of the LIVE/DEAD viability kit is called "SYBR", not "SYBER". Please correct this in the proofs.

Appendix A

Associate Editor Comments to Author (Dr Kristina Sefc):

Dear authors,

Your manuscript has been seen by three reviewers, who applaud your experiment and offer some advice for further improvement of your manuscript. Reviewer 1 comments on an important point, namely the simulation of natural heat waves, which involve fluctuations rather than stable temperature. This is a matter of what you claim to demonstrate in the manuscript, and can be resolved by revising your interpretation. When shortening the methods, please be careful to still include all information. Reviewer 1 also suggests that a 'read me first' text file should be added to the supplementary data, as the terms and headers in the tables are not comprehensible.

Reviewer 2 recommends cutting down the text of the discussion. Although there are no space limits, please keep in mind that readers will appreciate a concise text. Please attend to all of the reviewers' comments, and I look forward to seeing your revision.

RESPONSE: We have attended to all reviewers' comments and altered the manuscript accordingly. In particular we have suppressed the use of the term heatwave throughout, reduced the statistical analysis methods section, provided a read me first' text file for the supplementary data, and reduced the length of the discussion. We also have removed the figures and supplementary material and resubmit as separate files.

Reviewer comments to Author:

Reviewer: 1

Comments to the Author(s)

A. This study investigates on the impacts of exposure to high temperature on the biology of different life stage in *Tribolium*: survival, offspring production, testis size and spermatozoal recovery. One interesting aspects is about the time necessary for reproduction rate to recover following exposure to high temperature depending on the life stage that experienced the heat treatment. In general, the paper is well written, the methods are very clear and easily reproducible from the information provided.

The main problem I have is that the study applied treatments with constant temperature, at either 40 or 42°C, for 5 consecutive days. In the field, during heat waves, the temperature is never constant at so high temperature for so long (daily fluctuations still operate). Therefore, the study cannot conclude on the impacts of heatwave conditions. Instead, this study provides effects of high temperature on the biology of this species (which is still of interest). But this is thermal biology study and not anymore a study on the effect of heatwaves. The word heatwave should be suppress almost everywhere in the main text (the same error can be found in previous studies published by same and other authors).

RESPONSE: We agree of course that natural heatwaves will show thermal variation, so were initially careful throughout (and in other publications) to assert that this was an experimental study, and that our treatments exposed individuals to 'experimental heatwaves' or 'heatwave conditions'. We could not mimic exactly a natural heatwave, as many experts disagree on what even defines the conditions for a heatwave, and natural heatwaves will all be different. Our primary aim was to understand basic reproductive sensitivities at different life stages and their potential recovery in the extreme conditions that might be experienced through the increasing incidence of heatwaves. As the reviewer requests, we have suppressed the term 'heatwave' almost everywhere in the text, changing it to '*thermal stress*', '*heat stress*' or '*high temperature conditions*'.

Of note, we already had a paragraph in the Discussion considering ‘*What is the relevance of our study for insects exposed to heatwaves in nature?*’ where we discussed the relevance of our findings to natural variation in thermal regimes.

B. In addition, the control treatment consisted in an exposure to 30°C (constant, again). However, the optimal temperature is said to be around 35°C. Therefore, I wonder why the control was not fixed at 35°C ? That would increase the deviation between control and high temperature treatments, reflecting the extent to which a moderate increase in temperature is sufficient to generate high impacts (although this is not new). Indeed, at some point lower on the thermal performance curve, there should be a lower temperature at which the biological traits will show similar values than during the high temperature treatments, and one can wonder to what extent the 30°C is near this temperature equivalence.

RESPONSE: The optimal temperature for biological population productivity in *T. castaneum* is 35°C, but this temperature is not necessarily optimal for practical management in the laboratory in terms of development speed and food usage, so we (in common with most other researchers using this model) use 30°C as our ‘standard’ thermal regime. Others use 25°C to slow things down, but generation time is then ~doubled. To explain this in the manuscript, we refer to the authoritative ‘manual’ on *T. castaneum* management: Brown, S.J., Shippy, T.D., Miller, S., Bolognesi, R., Beeman, R.W., Lorenzen, M.D., Bucher, G., Wimmer, E.A., Klingler, M., 2009. The red flour beetle, *Tribolium castaneum* (Coleoptera): A model for studies of development and pest biology. *Cold Spring Harb. Protoc.* **4**, 1-9. doi:10.1101/pdb.emo126.

In addition, to be clearer about the ‘control’ 30°C temperature, we have added information to the methods section:

‘Thermal stress at either 40 (± 0.5)°C or 42 (± 0.5)°C temperatures were applied to beetles for 5 days at 60 (± 5)% RH, creating conditions that were 5°C to 7°C above the 35°C optimum for several life history traits and overall population fitness and productivity in T. castaneum (Sales et al., 2018b; Sokoloff, 1974), and 10 to 12°C above the standard experimental stock maintenance temperature of 30°C.’

‘Parallel controls were kept in standard identical conditions but at 30°C, which is a widely accepted experimental temperature regime for T. castaneum (e.g. Brown et al. 2009).’

C. The section on the statistical methodology is unusually long (4 pages) and should certainly be written in a more concise way by summarizing the first part on the general approach (the specific paragraphs after that provide the necessary details).

RESPONSE: We have cut back this section greatly as suggested, reducing the general approach to one paragraph, followed by the descriptions of the specific analysis approaches. We note that detailed analytical approaches are becoming a more common requirement in modern scientific publishing to ensure complete transparency about data handling.

Reviewer: 2

Comments to the Author(s)

A. I have read this paper a couple of times through and find the writing, analyses and interpretation are all clear. The work is well-justified, comprehensive and makes an excellent contribution to the growing literature, partially by this group, on better quantifying the potential role of climate-induced heat stress on insect reproduction.

RESPONSE: Thank you for the positive comments!

B. My only minor criticism is that the Discussion sometimes felt more like a review and if there are space considerations then I would recommend cutting down on line 779-852. The information in those sections are relevant but go somewhat beyond interpretation of the current results and future directions.

RESPONSE: We have shortened and tightened the Discussion between lines 779 and 852 (see Track Changes deletions). However, we retain some of the detailed discussion on 1) what might make sperm so sensitive, and 2) what might explain life stage sensitivity and recovery times, as these questions are of key relevance to our study's findings.

Reviewer: 3

Comments to the Author(s)

A. This is a beautifully written manuscript on the consequences of heatwaves on the mortality of different life stages and, among the survivors, on the downstream effects on reproductive traits and function, using red flour beetles as a model organism. The authors found severe effects of heat stress on post-larval stages, with virtually all pupae dying or, in early imagoes, becoming sterile (including disrupted testicular development and reduced sperm viability) – at least temporarily.

This paper describes a carefully designed and executed experiment testing well-motivated hypotheses on a timely and relevant topic. It must be the most clearly and eloquently written manuscript of the many I have already peer-reviewed this year and probably one of my first to accept almost as is. My comments, therefore, are mostly of editorial nature. Chapeau to the authors!

RESPONSE: Thank you very much for the very positive comments!

Here a few minor comments for further clarification:

B. Lines 77-78: What do you mean by “increases in hotter environmental temperatures”? Simply increases in environmental temperature, increases in temperature particularly in hot climates (i.e. no change in more polar regions), or increases in average temperatures particularly of the hot season (i.e. no change in winter)?

RESPONSE: We now expand this sentence to clarify, and state: *‘More recently, analyses across 66 bumble bee species throughout North America and Europe have revealed that extinctions or failed colonisations are specifically linked to “areas where local temperatures more frequently exceed species’ historical tolerances” (Soroye et al., 2020).’*

C. Line 270: Is this error around mean post-eclosion time the SD or SE? Please define at first mention. And sample size = 49?

RESPONSE: This is just a simple range value (i.e. most males were 31 days of age post eclosion, but a few were a minimum of 29 days, or a maximum of 33 days) which we have used throughout for temperatures and other age ranges. We can change further, but most readers should see this as a range figure, and hopefully not a standard deviation or error. We also now add in the combined total n = 49 sample size as suggested (but note that the n values for the three separate treatments are cited just above this sentence.)

D. Line 505 and throughout: It would be helpful to remind the reader of the sample sizes (or degrees of freedom) associated with all these z-statistics as it seems cumbersome to go find them in the Methods or supplementary material.

RESPONSE: We have cited the sample sizes throughout the methods, in all figures, and in the supplementary material, and now also provide them in the statistical Results section as requested.

E. Lines 505-507: Is this really such a coincidence that 3 p-values in a row are identical to three decimal places ($p = 0.671$) despite vastly different z-values, or were these results copy-pasted and accidentally not updated?

RESPONSE: Great spot – our copy-paste error - and now rectified; thank you.

F. Line 601: What do you mean by “immature, mature males and controls”? Immature and mature males relative to control males or immature, mature and control males with one another? Or what are immature, mature males if not two different sets of males?

RESPONSE: We have added information to remind readers that immature males are those exposed before adult maturity, which is reached at day 10 post eclosion, and expanded this sentence accordingly:

‘The overall comparison of reproductive recovery by immature males (exposure commenced within 1 day of adult eclosion) versus mature males (exposure commenced at 10 days following adult eclosion) and their respective controls following a single period of heat stress did not yield a significant difference between treatment groups ($\chi^2_{(2,42)} = 3.8, p = 0.159$; Figure 5b).’

G. Line 863: sable -> stable

RESPONSE: Thanks for the spot – now corrected.